# Risk Factors for Depressive Symptoms in Long-Haul Truck Drivers

**DOI:** 10.3390/ijerph17113764

**Published:** 2020-05-26

**Authors:** Alexander M. Crizzle, Maeve McLean, Jennifer Malkin

**Affiliations:** School of Public Health, University of Saskatchewan, Saskatoon, SK S7N 2Z4, Canada; maeve.mclean@usask.ca (M.M.); jnm040@usask.ca (J.M.)

**Keywords:** transportation, depression, mental health, fatigue, long-haul truck drivers

## Abstract

Work-related stress is a salient risk factor for depression. While long-haul truck drivers (LHTDs) face a myriad of occupational pressures and demands, little research has examined predictors of depressive symptoms in this occupational group. The purpose of this study was to identify predictors of depressive symptoms in LHTDs. A cross-sectional study was used to examine depressive symptoms, health and working conditions in a sample of 107 LHTDs (mean age of 50.7 ± 12.3; 95.6% were men) at truck stops from five Western Canadian cities. The findings show that 44% of LHTDs reported symptoms of depression in the past 12 months. Severe work-related stress, the use of psychiatric medications and broken sleep were significant predictors of depressive symptomology accounting for 41% of the variance. The findings suggest that LHTDs experience a host of occupational stressors that are embedded within the transportation industry that may increase the risk for depressive symptoms. Mental health promotion efforts that improve sleep quality, decrease work-related demands and pressures, and increase the use of psychiatric medication may reduce rates of depressive symptoms among LHTDs.

## 1. Introduction

Multiple studies show that chronic stress is a risk factor for the development of depressive symptoms [1,2,3]. Depressive symptoms refer to feelings of sadness, low self-esteem, psychological or emotional detachment, irritability and chronic fatigue that do not meet the official definition of depression according to the DSM-IV [4]. In occupational settings, excessive demands, extreme time pressures, low decision-making power, inadequate social support, and job insecurity have been linked with the development of depressive symptoms [5,6,7]. Chronic work-related stress can lead to increasing levels of neurotransmitters (e.g., epinephrine) and hormones (e.g., cortisol) [8,9] resulting in brain degeneration and depressive symptomology [10]. While these characteristics are endemic to the transportation sector, the association between stress and depression or depressive symptoms has not been extensively examined in long-haul truck drivers (LHTDs). LHTD refers to company drivers, lease owner-operators and independent owner-operators that drive a freight truck over thousands of kilometers, and do not return home for days or weeks at a time.

LHTDs face a myriad of occupational pressures and demands. The industry is characterized by long work hours, irregular work periods, social isolation, disrupted sleep schedules and extreme time pressures [11,12,13,14]. LHTDs report high rates of stress, anxiety, and burnout, all of which are risk factors for depression [15,16]. While qualitative studies have reported depression in LHTDs [15,17], only two studies have reported the prevalence rates of depressive symptoms, ranging from 19% to 26.9% [14,18]. One study found that depressive symptoms in truck drivers were predicted by younger age, poorer education, stimulant use, and being a company driver [15]. However, other risk factors that could contribute to depressive symptoms such as work-related stress, number of work hours, sleep quality, marijuana use, psychiatric medication use have yet to be examined. To date, there is limited information on what risk factors are associated with depressive symptoms in LHTD. Thus, the aim of the present study is to examine what risk factors are associated and predictive of depressive symptoms in LHTDs.

## 2. Materials and Methods

### 2.1. Data Collection and Procedures

Approval for this study was obtained from the University of Saskatchewan Research Ethics Board. Data were collected from four cities in Alberta (Lloydminster, Edmonton, Red Deer and Calgary) and one city in Saskatchewan (Saskatoon). Two research team members trained in survey methodology collected data in the evenings around the time of typical shift change so as to capture the perspectives of both day-shift and night-shift LHTDs. Permission was granted from truck stop managers to advertise and collect data at each truck stop location. LHTDs were approached and invited to participate in the study upon entering the truck stop. Written informed consent was obtained prior to study participation. Self-administered surveys were completed in a common area, often a restaurant or lounge. The survey took approximately 30–60 min to complete. At the end of the study, the truck drivers received a $20 honorarium for their participation. 

### 2.2. Participants 

Eligible participants included those who were (1) Canadian; (2) had a Class 1 driver’s license (or equivalent); and (3) had spent at least one night away from home while delivering a load at the time of recruitment. Individuals not meeting the inclusion criteria were excluded from the study. In total, 238 LHTDs participated in the study, however, only 107 (95.6% male; 4.4% female) provided information on depressive symptoms.

### 2.3. Survey

Occupational and health-related data were collected on LHTDs using a survey from a prior study in the USA [19]. This survey focused on factors which may impact the health and wellness of the LHTD, including topics such as health conditions, risk factors [20,21] and sleep [20]. This survey was adapted to a Canadian context. The survey was composed of several sections: health characteristics, driver and employment history, accident and work-related injuries, training, company safety climate/culture, and other safety-related questions. Depressive symptomology was assessed on the survey by asking participants whether they were depressed in the past year, as well as asking about distress or feelings of being upset in the past year. Participants were categorized as having depressive symptoms if they answered yes to feeling depressed in the past year.

### 2.4. Data Analysis

Surveys were entered into SPSS version 25 for analysis. Sample characteristics were examined using descriptives (Mean ± SD; and range) or frequencies (valid percentages). Prior to inferential statistics, variables were assessed for normality. Correlations (Pearson r or Spearman rho), chi-squares, and independent t-tests (or Mann−Whitney U) were performed to examine the relationship between work-related demands, stress, health status, substance use, sleep quality, and fatigue with depressive symptoms. Variables significantly associated with depressive symptoms were entered into a stepwise backward logistic regression model to determine predictors of depressive symptoms.

## 3. Results

### 3.1. Sample Demographics

The sample ranged in age from 24 to 89 years (M = 50.7 ± 12.3). Most drivers identified as Caucasian (82.6%) with the remaining drivers identifying as Asian (5.8%), Black or African American (2.9%), Latino/Hispanic (1.9%), East Indian (2.9%), American Indian or Alaskan Native (1%), and other (2.9%). Participants were primarily company employees (70%) while 26% were owner-operators under a lease agreement and 4% were independent owner-operators. The number of years worked as an LHTD ranged from 6 months to 56 years (M = 21.8 years ± 13.9).

### 3.2. Associations with Depressive Symptoms

As shown in Table 1, 44% of LHTDs reported depressive symptoms in the past twelve months. Truck drivers who reported depressive symptoms were significantly more likely to report severe work stress (*p* = 0.007), marijuana use (*p* = 0.020), low back pain (*p* = 0.004), fewer hours of sleep per night (*p* = 0.003), broken sleep (*p* < 0.001), sleep apnea (*p* = 0.032), the use of a CPAP machine (*p* = 0.020), more frequent weekly fatigue (*p* = 0.007), higher levels of daily fatigue (*p* = 0.001), prior mental health treatment (*p* = 0.022), and the use of psychiatric medications (*p* = 0.040). Associations between depressive symptoms with annual income and weeks worked approached significance (both *p* = 0.054, respectively).

### 3.3. Predictors for Depressive Symptoms

As shown in Table 2, all significant variables were inputted into a backward logistic regression, which generated eight models. Variables contributing the least to each model were continuously removed until a final model was produced. The first model included ten variables that accounted for 44% of the variance. The final model included four variables, three of which were significant predictors of depressive symptomology, and accounted for 40.7% of the variance. The final model (*n* = 61, −2 log likelihood ratio = 60.619; Nagelkerke R^2^ = 0.407) found that severe work-related stress (*p* = 0.045), the use of psychiatric medications (*p* = 0.034), and broken sleep (*p* = 0.017) were significant predictors of depressive symptomatology in LHTD. 

As shown in Table 3, the odds of having depressive symptoms were 3.778 times greater for those with severe work-related stress (CI = 1.031, 13.845), 5.148 times greater for those with broken sleep (CI = 1.552, 20.598) and 16.343 times greater for truck drivers who used psychiatric medications (CI = 1.343, 216.817).

## 4. Discussion

The findings indicate that severe work-related stress, broken sleep (e.g., poorer quality sleep), and the use of psychiatric medications were predictive of depressive symptoms in LHTDs. Drivers who reported severe work-related stress were almost four times more likely to report having been depressed in the past year. While prior studies have not examined work-related stress directly, work-related factors such as long work hours, constant time pressures, social isolation, poor driving conditions, fear of violence, and low levels of job satisfaction and control are associated with psychological strain and emotional distress [14,15,17,22] Moreover, research suggests that social stressors, particularly social isolation and low social status, activate brain regions that are responsible for producing defensive, physiological responses, such as increasing levels of cortisol and pro-inflammatory cytokines. In turn, the increasing levels of cortisol induce the neurobiological and behavioral changes that characterize depressive disorders [23]. These findings suggest that occupational stressors that are endemic to the LHTD sector [12,15] may put drivers at risk for depressive symptoms. 

In the current study, LHTDs who reported broken sleep were more than five times as likely to have reported being depressed in the past year. A prior study similarly found that truck drivers who reported poor sleep quality were 2.58 times more likely to be psychologically distressed [24] with another study finding that insomnia was associated with depression [1]. Studies showed that excessive work demands, in combination with night shifts, irregular and long daily schedules, led to poor sleep habits and sleep disturbances in truck drivers [25]. Other studies have found that truck drivers reported poor sleep quality, short sleep duration, insomnia, and excessive daytime sleepiness due to irregular work-rest schedules [1,26,27]. Moreover, prior studies showed that truck drivers were significantly more likely to report sleep disorders when compared to other occupational groups [28]. Research shows that poor sleep quality leads to disruptions in emotional regulation due to impairments to the pre-frontal cortex and the functioning of the limbic system [29]. These findings suggest that poor sleep quality may significantly contribute to the development, exacerbation, and relapse of depressive symptomology among LHTDs. 

Depressive symptomology was also predicted by psychiatric medication use. Drivers who reported psychiatric medication use were 16 times more likely to have reported depressive symptoms in the past year. However, 85% of drivers who reported depressive symptoms had not received any form of psychiatric medication and 80% had not received any professional mental health treatment in the past year. This is consistent with research conducted by Shattell and colleagues which found that 91.6% of truck drivers did not receive prescription medication for mental health problems [14]. Prior studies showed that truck drivers reported low levels of healthcare utilization due to interprovincial travel, irregular work schedules, a lack of health care insurance, and an inability to take medical leave from their transport company employers [14,30]. In the current study, 43% of LHTD with depressive symptoms did not have any employment-based health benefits. Therefore, low levels of psychiatric medication use may be due to increasing work demands and the lack of employee benefits within the transportation sector [12,15]. 

The prevalence of depressive symptoms in the past year was 44% in our sample of LHTDs. Prior studies have reported rates ranging from 19% in the past month [18], to a lifetime prevalence of 27% in LHTDs [14]. Another study found that 13.6% were depressed, based on a clinical diagnostic interview and questionnaire [16]. Studies have asked about depressive symptoms over the course of a month [18], over the past year (present study), and over a lifetime [14]. Only one study reported a clinical diagnosis of depression in a LHTD [16]. Additionally, although Shattell and colleagues [14] asked about depression over lifetime, their sample was significantly younger (mean age of 37.8 vs. 50.7 in the present study) and had lessworking experience than the present sample, perhaps explaining why our depressive symptom rates are higher compared to prior studies.

This study determined predictors of depressive symptoms in LHTDs. However, there are several study limitations. The study was cross-sectional in nature. Although we examined associations between independent variables and depressive symptoms, causality cannot be determined. Additionally, we cannot determine whether the predictors outlined in this study are unique to the LHTD, since the current study did not examine predictors of depression between other occupational groups in Canada. Moreover, recruitment was based on convenience sampling and we cannot infer that our findings are generalizable to the broader LHTD population. Furthermore, the specific type of psychiatric medication was not collected. While antidepressants are one type of psychiatric medication, other types of psychiatric medications include anti-anxiety medication, anti-psychotics or mood stabilizers. And lastly, we relied on self-reported data which is subject to social desirability bias. Longitudinal studies are needed to determine what risk factors are prospectively predictive of depression in LHTDs.

## 5. Conclusions

LHTDs experience a host of occupational stressors that are embedded within the transportation environment that put drivers at risk for depression. To determine if depressive symptoms are endemic to LHTDs, future research should consider comparing rates and risk factors for depressive symptoms with other non-LHTD occupations. This could result in tailored mental health promotion efforts within the broader context of the transportation industry to reduce the high rates of depressive symptoms in the trucking industry. Efforts made to improve sleep quality, decrease work-related demands and pressures, and increase the use of psychiatric medication among LHTDs could potentially improve the rates of depressive symptoms. Possible avenues for addressing mental health in LHTDs include educational programs, online support groups and telehealth services, which to date, have not yet been examined.

## Figures and Tables

**Table 1 ijerph-17-03764-t001:** Sample characteristics and comparisons by group.

Sample Characteristics	Depressive Symptoms *n* = 47	No Depressive Symptoms *n* = 60	Sig.
Age ^2^	49.3 ± 10.9 range 24–70	51.9 ± 13.4 range 25–89	0.323
Gender ^1^			
Male	98%	94%	0.621
Female	2%	6%	
Ethnicity ^1^			
Caucasian	86%	80%	0.090
Black or African American	0%	5%	
Asian	7%	5%	
Native American Indian	0%	2%	
Pacific Islander	0%	0%	
Latino/Hispanic	0%	3%	
East Indian	0%	5%	
Other	0%	0%	
Level of education ^1^			
Not completed high school	39%	43%	0.774
High school diploma	26%	26%	
Some college	22%	17%	
Completed post-secondary	13%	14%	
Marital status ^1^			
Single	17%	18%	0.442
Common law	20%	10%	
Married	35%	47%	
Divorced	28%	25%	
Annual income ^2^	$75,138 ± 26,494 range 35,000–180,000	$106,235 ± 86,869 range 35,000–450,000	0.054
Employment type ^1^			
Company employee	78%	64%	0.510
Lease owner-operator	20%	31%	
Independent owner-operator	2%	5%	
Kilometres per year ^2^	172,374 ± 69,628 range 25,000–340,000	175,017 ± 75,179 range 3500–360,000	0.537
Weeks worked per year ^2^	48.738 ± 5.142 range 30–52	45.564 ± 12.301 range 4–52	0.088
Hours worked per week ^2^	63.195 ± 29.501 range 11–143	55.713 ± 28.269 range 3–100	0.190
Days slept at home per month ^2^	3.0 ± 19.7 range 0–30	2.9 ± 13.2 range 0–25	0.541
Percentage of night-time driving ^2^	25.3 ± 22.5 range 0–80	27.0 ± 22.8 range 0–100	0.704
Electronic log-book use ^1^			
Yes	41%	49%	0.450
No	59%	51%	
Employment-based health insurance ^1^ Yes No	57% 43%	66% 34%	0.245
Severe job-related stress ^1^			
Yes	62%	34%	**0.005**
No	38%	66%	
Mental health consultation ^1^ (lifetime prevalence)			
Yes	68%	17%	**0.001**
No	32%	83%	
Mental health treatment ^1^ (past year)			
Yes	20%	5%	**0.022**
No	80%	95%	
Use of psychiatric mediation ^1^ (past year)			
Yes	15%	3%	**0.040**
No	85%	97%	
Alcohol consumption ^2^ (days/month)	0.63 ± 1.21 Range 0–4	0.77 ± 1.99 range 0–10	0.828
Alcohol consumption ^1^ (drinks/day)			
None	79%	69%	0.684
1–2	15%	22%	
3–4	6%	7%	
5–6	0%	2%	
7–9	0%	0%	
10 or more	0%	0%	
Cigarette smoking ^1^			
Every day	46%	43%	0.774
Some days	10%	7%	
Not at all	44%	50%	
Marijuana use ^1^			
Yes	16%	2%	**0.020**
No	84%	98%	
Marijuana use per day ^2^	0.20 ± 0.80 range 0–4	0.10 ± 0.78 range 0–6	0.170
Self-reported health rating ^1^			
Excellent/very good	16%	29%	0.088
Good	50%	54%	
Fair/poor	34%	17%	
Days of physical activity/week ^2^	3.0 ± 2.3 range 0–7	2.8 ± 2.6 range 0–7	0.638
Healthy diet ^1^			
Yes	66%	70%	0.236
No	44%	30%	
Low back pain ^1^			
Yes	52%	23%	**0.004**
No	48%	77%	
Hours of sleep/night ^2^	6.8 ± 1.8 range 4–15	7.6 ± 1.6 range 2–10	**0.003**
Sleep quality ^1^			
Broken	79%	35%	**0.000**
Continuous	21%	65%	
Sleep apnea ^1^			
Yes	25%	8%	**0.032**
No	75%	92%	
CPAP machine use ^1^			
Yes	16%	2%	**0.020**
No	84%	98%	
Fatigue frequency (days/week) ^2^	2.6 ± 2.5 range 0–7	1.6 ± 2.3 range 0–9	**0.007**
Fatigue severity (0–10)	2.7 ± 2.2 range 0–8	1.6 ± 2.2 range 0–10	**0.001**
Drowsiness while driving ^1^			
Never	30%	44%	0.135
Once/month	7%	15%	
At least once/week	43%	29%	
Almost every day	20%	12%	
Number of at-fault crashes ^2^	0.27 ± 0.59 range 0–2	1.7 ± 6.5 range 0–30	0.389

^1^*p*-value was computed using a Pearson’s chi squared test, ^2^*p*-value was computed using an independent samples *t*-test. Bold values indicate statistical significance.

**Table 2 ijerph-17-03764-t002:** Predictors of self-reported depressive symptoms.

Entered Variables	Model 1 R^2^ = 0.440	Model 2 R^2^ = 0.440	Model 3 R^2^ = 0.439	Model 4 R^2^ = 0.438	Model 5 R^2^ = 0.437	Model 6 R^2^ = 0.432	Model 7 R^2^ = 0.422	Model 8 R^2^ = 0.407
Severe job-related stress	0.160	0.153	0.156	0.129	0.138	0.104	0.044	0.045
Psychiatric Medication use	0.170	0.169	0.164	0.092	0.100	0.100	0.022	0.034
Broken sleep	0.020	0.020	0.020	0.019	0.020	0.020	0.022	0.017
CPAP machine use	0.244	0.245	0.170	0.177	0.184	0.130	0.119	0.104
Hours of sleep/night	0.401	0.405	0.370	0.323	0.315	0.334	0.324	
Mental health treatment	0.367	0.365	0.372	0.389	0.386	0.422		
Marijuana use	0.591	0.593	0.588	0.556	0.586			
Low back pain	0.705	0.696	0.721	0.728				
Fatigue frequency (days/week)	0.785	0.787	0.796					
Sleep apnea	0.807	0.810						
Fatigue severity (0–10)	0.918							

**Table 3 ijerph-17-03764-t003:** Regression coefficients.

Predictor	β	Sig.	Odds Ratio	95% CI
Severe job-related stress	1.329	0.045	3.778	1.031, 13.845
Psychiatric medication use	2.794	0.034	16.343	1.232, 216.817
Broken sleep	1.639	0.017	5.148	1.345, 19.711

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
