# Peer review of "Risk Factors for Depressive Symptoms in Long-Haul Truck Drivers"

_ijerph, 2020, doi:10.3390/ijerph17113764_

Round 1

Reviewer 1 Report

The manuscript (ms) describes a survey study whereby a convenience sample of long-haul truck drivers (LHTD) were recruited and polled at truck stop locations in Alberta and Saskatoon, Canada. The purpose of the study was to identify the risk factors associated depressive symptoms in LHTD.

The ms is well written and clearly presented. My only concern is with respect to the design, the lack of consideration for non-LHTD, and the resulting conclusions that the authors make (given the study design). Put simply, how do the authors know that the factors being assessed are unique to LHTD? That is, could the factors identified be relevant to all blue-collar workers? All workers? All Canadians? Etc? To make conclusions for LHTD, I think the authors need a design that teases out factors that may similarly occur in a non-LHTD cohort. I’ve included a reference below that the authors might consider that similarly dealt with health-related factors in truck drivers.

I would recommend that the study be expanded include a cohort of non-LHTD workers so a clearer picture can be gained specifically for truck drivers (which is the stated goal). Short of that, the authors should address this as a study limitation.

Thiese, M. S., Hanowski, R. J., Moffitt, G., Kales, S.N., Porter, R.J., Ronna, B., Hartenbaum, N., Hegmann, K.T. (2018). A retrospective analysis of cardiometabolic health in a large cohort of truck drivers compared to the American working population. American Journal of Industrial Medicine. doi:10.1002/ajim.22795.

Author Response

Response to the Editor’s and Reviewers’ Comments: May 10th, 2020

Manuscript #: IJERPH-808995_R1  

Title of Paper: Risk factors for depressive symptoms in long-haul truck drivers

Thank you for the review of our manuscript and the invitation to address the reviewers’ comments and revise our article. We have made numerous changes to strengthen the manuscript based on comments provided. Below, we have indicated how we have addressed the reviewer’s comments.

Reviewer 1 Comments:

The manuscript (ms) describes a survey study whereby a convenience sample of long-haul truck drivers (LHTD) were recruited and polled at truck stop locations in Alberta and Saskatoon, Canada. The purpose of the study was to identify the risk factors associated depressive symptoms in LHTD.

The ms is well written and clearly presented. My only concern is with respect to the design, the lack of consideration for non-LHTD, and the resulting conclusions that the authors make (given the study design). Put simply, how do the authors know that the factors being assessed are unique to LHTD? That is, could the factors identified be relevant to all blue-collar workers? All workers? All Canadians? Etc? To make conclusions for LHTD, I think the authors need a design that teases out factors that may similarly occur in a non-LHTD cohort. I’ve included a reference below that the authors might consider that similarly dealt with health-related factors in truck drivers.

I would recommend that the study be expanded include a cohort of non-LHTD workers so a clearer picture can be gained specifically for truck drivers (which is the stated goal). Short of that, the authors should address this as a study limitation.

Thiese, M. S., Hanowski, R. J., Moffitt, G., Kales, S.N., Porter, R.J., Ronna, B., Hartenbaum, N., Hegmann, K.T. (2018). A retrospective analysis of cardiometabolic health in a large cohort of truck drivers compared to the American working population. American Journal of Industrial Medicine. doi:10.1002/ajim.22795.

Response: Thank-you for the comment. We have now included the recommended reference in the discussion section, as well as mentioned the inability to compare depression rates of truck drivers to other working classes as a limitation and an opportunity for further research. Specific changes are shown below.

Line #135: We have also revised the following sentence: “These findings suggest that occupational stressors that are endemic to the LHTD sector [9,12] may put drivers at risk for depressive symptoms.

Line 145: The following sentence was added:

“Moreover, prior studies show that truck drivers are significantly more likely to report sleep disorders when compared to other occupational groups [25].”

Line #152: Added the following sentence: “Additionally, we cannot determine whether the predictors outlined in this study are unique to LHTD since the current study did not examine predictors of depression between other occupational groups in Canada.”

Reviewer 2 Report

- abstract: inform the type of study, sex and number of participants. Is it not necessary to inform the tests carried out, were truckers on urban, road or rural routes? Check if the Keywords are in the MESH. Don't just drop suggestions to complete in response to the goal.

- introduction
Although the introduction briefly addresses depression and the target audience, it nevertheless makes no reference to the different types of truck drivers and does not delve further into explaining the physiological relationship between work and stress, which is suggested.

- methods
Indicate the type of study. How many researchers carried out the evaluations? Were they trained? It occurred during which time of day? Night? Evening? Morning?
I suggest not indicating the payment of remuneration to volunteers.
What are the exclusion criteria?
Has the collection instrument been validated? Was there an internal confidence analysis, pre-test, or some form of validation of the questionnaire for the public and parents in which the study was conducted?
The analysis of depression requires a better understanding of the questions in the questionnaire, given that this disease is diagnosed by a doctor or psychologist, did you have any on the team? Were the questions that initially diagnosed depression elaborated, validated, tested by a doctor or psychologist?
There are specific validated questionnaires for quality of life, sleep and depression, why didn't you use them?

- Results
Put 0 (zero) before, in the value of p
Indicate in each table, in the legend, the test used for each analysis in all tables
Indicate what is the requirement to insert the variables in the models, p <0.2 in the chi?

- discussion
It is to be desired when it does not justify, based on the literature, the findings, just comparing with what already exists does not bring desirable scientific reflections. It is suggested that when presenting each finding, seek to justify with bibliographic / physiologically / biologically based how each finding influences the body systems and hence the cascade of events.
In general, the discussion needs the above reports.

Author Response

Response to the Editor’s and Reviewers’ Comments: May 10th, 2020

Manuscript #: IJERPH-808995_R1  

Title of Paper: Risk factors for depressive symptoms in long-haul truck drivers

Thank you for the review of our manuscript and the invitation to address the reviewers’ comments and revise our article. We have made numerous changes to strengthen the manuscript based on comments provided. Below, we have indicated how we have addressed the reviewer’s comments.

Reviewer 2 Comments:

- abstract: inform the type of study, sex and number of participants. Is it not necessary to inform the tests carried out, were truckers on urban, road or rural routes? Check if the Keywords are in the MESH. Don't just drop suggestions to complete in response to the goal.

Response: We have revised the abstract to include information on the study design, as well as the number of participants and their basic demographics (age, gender). We made the following revision below:

Line # 15: Revised the following sentence: “A cross sectional study was used to examine depressive symptoms, health and working conditions in a sample of 107 LHTD (mean age of 50.7±12.3; 95.6% were men) at truck stops in five Western Canadian cities”  

Line #17: Revised the following sentence: “Severe work-related stress, the use of psychiatric medications and broken sleep were significant predictors of depressive symptomology accounting for 41% of the variance.”

Line #26: MESH Terms: As stated in the journal guidelines, 3-10 key works are to be provided that are specific to the article “Three to ten pertinent keywords need to be added after the abstract. We recommend that the keywords are specific to the article, yet reasonably common within the subject discipline”. However, we have inserted MESH key words with the exception of long-haul truck drivers, since no MESH words are representative of this segment of the driver population.  

Introduction:

Although the introduction briefly addresses depression and the target audience, it nevertheless makes no reference to the different types of truck drivers and does not delve further into explaining the physiological relationship between work and stress, which is suggested.

Response: There are numbers truck driver professions including dump truck driving, tow truck driving, flatbed truck driving, heavy weight truck driver, etc. Therefore, it is not feasible to outline all types of truck driving professions. And this article specifically examined long-haul truck drivers, considered the most at-risk for the development of chronic medical conditions. We have revised the last sentence of the introduction to reiterate that we are examining long-haul truck drivers, as well as included a definition of long-haul truck drivers.

Line #41: Added the following sentence: “LHTD refer to company drivers, leased owner-operators, and independent owner-operators that drive a freight truck over thousands of kilometers and do not return home for days or weeks at a time.”

We have also revised the introduction and added a sentence denoting the link between the physiological relationship between stress and depression.  

Line #34: The following sentence was added: “Chronic work-related stress can lead to increasing levels of neurotransmitters (e.g. epinephrine) and hormones (e.g. cortisol) [8,9] resulting in brain degeneration and depressive symptomology [10].”

Methods:

Indicate the type of study.

Response: We have indicated the type of study: specified that it was cross-sectional study (Line 101)

How many researchers carried out the evaluations? Were they trained? It occurred during which time of day? Night? Evening? Morning?

Response: We added that data were collected by two trained research members (Line 103-4). We also added that data were collected in the evenings around the time of typical shift change to capture both day-shift and night-shift LHTD (Line 104-5).

I suggest not indicating the payment of remuneration to volunteers.

Response: We did not remove the fact that participants were given a $20 honorarium for participating as this is a part of our methods. Additionally, this is standard to note in research studies due to potential issues related to bias (see discussion section).

What are the exclusion criteria?

Response: We have clarified the exclusion criteria and added the following sentence: “LHTD who did meet these inclusion criteria were excluded from the study.”

Has the collection instrument been validated? Was there an internal confidence analysis, pre-test, or some form of validation of the questionnaire for the public and parents in which the study was conducted? There are specific validated questionnaires for quality of life, sleep and depression, why didn't you use them?

Response: Concerning the survey, it was developed by the National Institute of Occupational Safety and Health (Sieber et al. 2014). Many of the survey instruments were previously validated including health conditions and risk factors (National Health Interview Survey (NCHS, 2006); CDC’s Behavioral Risk Factor Surveillance System (CDC, 2007)) and sleep (NCHS, 2006). We have added a sentence to the methods mentioned these details (line 119-121). Additionally, the survey itself was almost 50 pages and to try and reduce burden and drop-outs, we did not want to include any longer scales in the existing survey.

The analysis of depression requires a better understanding of the questions in the questionnaire, given that this disease is diagnosed by a doctor or psychologist, did you have any on the team? Were the questions that initially diagnosed depression elaborated, validated, tested by a doctor or psychologist?

Response: We did not have a doctor or psychologist on the team; however, we were analyzing depressive symptoms based on what is defined in the DSM-IV rather than attempting to provide a clinical diagnosis of depression.

Results

Put 0 (zero) before, in the value of p

Response: We have now placed a “0” in front of all p-values in the manuscript and corresponding tables.

Indicate in each table, in the legend, the test used for each analysis in all tables

Response: We have provided a footnote indicating what test was conducted in Table 1 (see Line 114). Details on the other tables are already provided in the text.

Indicate what is the requirement to insert the variables in the models, p <0.2 in the chi?

Response: We have stated that all significant variables were included in the logistic regression model (Line 116).  

Discussion

It is to be desired when it does not justify, based on the literature, the findings, just comparing with what already exists does not bring desirable scientific reflections. It is suggested that when presenting each finding, seek to justify with bibliographic / physiologically / biologically based how each finding influences the body systems and hence the cascade of events.
In general, the discussion needs the above reports.

Response: Thank-you for your comment. We have added a few sentences further mentioning the link between our findings with depression. For example, we have now inserted a sentence stating:

“Moreover, research suggests that social stressors, particularly social isolation and low social status, activate brain regions that are responsible for producing defensive, physiological responses, such as increasing levels of cortisol and pro-inflammatory cytokines. In turn, the increasing levels of cortisol induces the neurobiological and behavioural changes that characterized depressive disorders [20].” – Line 152

We have also added the following sentence: “Research shows that poor sleep quality leads to disruptions in emotional regulation due to impairments in pre-frontal cortex and limbic system functioning [25].” – Line 154

Round 2

Reviewer 1 Report

Thank you to the authors for revising the ms and addressing my concerns. Based on the study design, and lack of a non-LHTD cohort, the value of the study is limited. Nonetheless, I think this paper could serve as an initial/pilot study, but a follow-on, with a better design, is strongly recommended. This notion could be included in the conclusions as a need for further research and next steps.  

Author Response

Thank-you for the additional suggestion. 

Response: We have now incorporated a sentence in the conclusion section suggesting future research could examine the rates and risk factors for depressive symptoms in LHTD to other occupations to determine if this finding is truly endemic to LHTD. 

Reviewer 2 Report

I dont have any coments

Author Response

Thank-you for your helpful suggestions for improving the manuscript.